# Gibberellin Signaling Promotes the Secondary Growth of Storage Roots in *Panax ginseng*

**DOI:** 10.3390/ijms22168694

**Published:** 2021-08-13

**Authors:** Chang Pyo Hong, Jinsoo Kim, Jinsu Lee, Seung-il Yoo, Wonsil Bae, Kyoung Rok Geem, Jin Yu, Inbae Jang, Ick Hyun Jo, Hyunwoo Cho, Donghwan Shim, Hojin Ryu

**Affiliations:** 1Theragen Bio Co., Ltd., Suwon 16229, Korea; seungil.yoo@theragenbio.com; 2Department of Biology, Chungbuk National University, Cheongju 28644, Korea; iekee1004@gmail.com (J.K.); jinsulee90@gmail.com (J.L.); plittle80@naver.com (W.B.); roki2012@naver.com (K.R.G.); 3School of Biological Sciences, Seoul National University, Seoul 08826, Korea; 4Department of Herbal Crop Research, National Institute of Horticultural and Herbal Science, Rural Development Administration, Eumseong 27709, Korea; yugin8603@korea.kr (J.Y.); ikanet@korea.kr (I.J.); intron28@gmail.com (I.H.J.); 5Department of Industrial Plant Science and Technology, Chungbuk National University, Cheongju 28644, Korea; russelcho@gmail.com; 6Department of Biological Sciences, Chungnam National University, Daejeon 34134, Korea; dshim104@cnu.ac.kr; 7Department of Biological Sciences and Biotechnology, Chungbuk National University, Cheongju 28644, Korea

**Keywords:** gibberellins, *Panax ginseng*, GID1s, phytohormones, storage root secondary growth, cell wall biogenesis

## Abstract

Gibberellins (GAs) are an important group of phytohormones associated with diverse growth and developmental processes, including cell elongation, seed germination, and secondary growth. Recent genomic and genetic analyses have advanced our knowledge of GA signaling pathways and related genes in model plant species. However, functional genomics analyses of GA signaling pathways in *Panax ginseng*, a perennial herb, have rarely been carried out, despite its well-known economical and medicinal importance. Here, we conducted functional characterization of GA receptors and investigated their physiological roles in the secondary growth of *P. ginseng* storage roots. We found that the physiological and genetic functions of *P. ginseng* gibberellin-insensitive dwarf1s (PgGID1s) have been evolutionarily conserved. Additionally, the essential domains and residues in the primary protein structure for interaction with active GAs and DELLA proteins are well-conserved. Overexpression of *PgGID1s* in *Arabidopsis* completely restored the GA deficient phenotype of the *Arabidopsis gid1a gid1c* (*atgid1a/c*) double mutant. Exogenous GA treatment greatly enhanced the secondary growth of tap roots; however, paclobutrazol (PCZ), a GA biosynthetic inhibitor, reduced root growth in *P. ginseng*. Transcriptome profiling of *P. ginseng* roots revealed that GA-induced root secondary growth is closely associated with cell wall biogenesis, the cell cycle, the jasmonic acid (JA) response, and nitrate assimilation, suggesting that a transcriptional network regulate root secondary growth in *P. ginseng*. These results provide novel insights into the mechanism controlling secondary root growth in *P. ginseng*.

## 1. Introduction

Korean ginseng (*Panax ginseng* C.A. Meyer) has been used as an important medicinal plant species for thousands of years in Asia, especially in Korea, Japan, and China [1]. Ginseng has the ability to increase immunity and vitality and to prevent aging, and its pharmacological effects have been proven through clinical trials and animal experiments [2]. Studies on ginseng have focused primarily on its pharmacological effects in humans; however, its unique physiological and developmental characteristics have rarely been explored thoroughly due to its perennial heterozygous nature [3]. The application of functional genomic approaches in ginseng has been challenging because of its unique physiological and ecological characteristics and the lack of genomic information [4,5], which could be attributed to its relatively large genome (3.2 Gbp), heterozygous allotetraploid nature (2*n* = 4x = 48), and significant amounts of repetitive DNA [6,7].

Plants, as sessile organisms, induce a variety of physiological changes in response to adverse environmental conditions. Plant hormones function as master regulators of growth and development under unsuitable external environments [8]. Gibberellins (GAs) are tetracyclic diterpenoid hormones that play essential roles in seed germination, flowering, pollen maturation, stress tolerance, secondary growth, and development [9,10,11]. Recent studies have shown that there are complicated internal and external signaling interactions of GAs and various hormone pathways, including auxin, abscisic acid (ABA), jasmonic acid (JA), ethylene, and cytokinin. These interactions are highly relevant to plant growth, development, and stress tolerance [12,13,14]. Major GA signaling components have been identified through genetic studies on model plant species, including rice (*Oryza sativa*) and *Arabidopsis thaliana*. The nucleocytoplasmic GA receptor gibberellin-insensitive dwarf1 (GID1) was first identified in rice through studies on gibberellin-insensitive dwarf mutants [15]. OsGID1 and AtGID1s have been characterized as water-soluble GA receptors that exhibit strong binding affinity for bio-active GAs [15,16]. At the C-terminus, GID1 harbors a well-conserved α/β-hydrolase domain, which belongs to the carboxylesterase family of plant proteins [17,18]. The canonical GA signaling pathway is initiated by direct binding of active GAs to GID1 in the nucleus. The GA–GID1 complex induces rapid degradation of the major growth-inhibitory factor DELLA through direct protein–protein interactions [12,19]. DELLAs are key transcriptional repressors that inhibit GA signaling and plant growth. Degradation of DELLA proteins requires both GID1 and the F-box protein SLY1 (SLEEPY)/GID2 [15,20,21,22]. In the presence of GA, DELLA is recognized by the SCF^SLY1/GID2^ ubiquitin E3 ligase complex for subsequent 26S proteasome-meditated degradation [21,23,24]. This canonical GA signaling pathway is responsible for various physiological responses during plant growth and development.

The secondary growth of roots is particularly important for production of various root crops and for accumulation of useful compounds and nutrients in storage roots [25,26,27]. Studies on root crops such as radish and cassava show that the secondary growth of storage roots is regulated by the activity of cambium stem cells. Secondary growth of plants involves formation of cambium stem cells and differentiation of phloem and xylem tissues, which are mainly regulated by the plant hormones auxin and cytokinin [25,26,27,28,29]. In addition, the role of GA in secondary xylem formation and lignification in storage roots has also been confirmed in several plant species, including poplar, carrot, and cotton [30,31,32,33]. Overproduction of GA promotes elongation and division of xylem and fiber cells in the vascular bundle and increases cambium activity [26,32,34]. However, in carrot and sweet potato, exogenous GA treatment inhibits root growth by affecting cell division and vascular lignin synthesis [35,36]. These results suggest that the physiological response to GA, a growth-promoting hormone, varies with the plant species. Although Korean ginseng is one of the most important medicinal root crops with a 6-year cultivation period, genetic and physiological factors affecting the growth and development of ginseng roots have rarely been investigated.

Owing to recent advances in sequencing technology, the draft genome sequence of *P. ginseng* has been released by two research groups [37,38]. In addition, we also previously reported the precise transcript sequences of *P. ginseng* using PacBio single-molecule real-time (SMRT) isoform sequencing (Iso-seq) analysis [39]. Herein, we demonstrate for the first time that GA promotes storage root secondary growth in *P. ginseng*. We identified eight putative homologs of the *AtGID1* gene in *P. ginseng* (*PgGID1s*), based on its draft genome sequence and SMRT Iso-seq data [37,38,39]. Functional genomic analysis of four representative *PgGID1s* (*PgGID1A–D*) revealed that the functions of PgGIDs are evolutionarily conserved. Transcriptome analysis further supported that GA-induced root secondary growth is tightly connected with cell wall biogenesis, the cell cycle, the JA response, and nitrate assimilation. These results are expected to facilitate development of an omics-based breeding technology for ginseng.

## 2. Results

### 2.1. GA Enhances Shoot Primary Growth and Root Secondary Growth in P. ginseng

The secondary growth of the tap root, as a storage organ, is one of the main factors affecting *ginseng* yield. To investigate the physiological effects of GA in *P. ginseng,* active GA_3_ and paclobutrazol (PCZ), a GA biosynthetic inhibitor, were applied exogenously to the roots of 1-year-old *P. ginseng* plants. Exogenous application of GA significantly promoted stem growth compared with mock treatment (Figure 1A,B), which is consistent with the role of GA as a growth-promoting hormone [40]. Interestingly, exogenous GA treatment also increased the diameter of the tap root; however, PCZ application reduced both shoot and root growth in *P. ginseng* (Figure 1A,B). To further analyze the GA-mediated promotion of shoot and root growth, GA- and PCZ-treated *P. ginseng* samples were applied to histological paraffin-embedded sections with Safranine-Astra blue combination staining. Compared with the control, plants in the GA treatment group showed longer epidermal cells, consistent with their elongated stem phenotype (Figure 1C). Conversely, the stems of plants in the PCZ treatment group showed slightly shorter cells compared with the control (Figure 1C). However, there was no significant difference in the size of the divided cells surrounding the cambium layer of the storage tap root, although the number of divided starch-deposited storage parenchyma cells located between xylem vessels and resin duct cells was greatly increased in GA-treated *P. ginseng* roots (Figure 1D). Consistently, the number of cambium-derived storage parenchyma and vascular cells in PCZ-treated roots was significantly reduced (Figure 1D). These results indicate that GA facilitates root secondary growth in *P. ginseng* by promoting storage parenchyma cell development.

### 2.2. Identification of Putative GA Receptor Genes in P. ginseng Genome

The growth-promoting effect of GA on the stem and root growth of *P. ginseng* suggests that the canonical GA signaling pathways are likely evolutionarily conserved in *P. ginseng*. Since GA signal transduction initiates with its perception, we searched for GA receptor-encoding genes in the *P. ginseng* genome by analyzing the two genome sequence drafts of *P. ginseng* and PacBio Iso-seq data [21,22,23]. A BLAST search of the AtGID1A amino acid sequence led to identification of eight putative *GID1* DNA sequences in *P. ginseng*, designated as *PgGID1A–H* (Figure 2A). Phylogenetic analysis of PgGID1s and AtGID1s showed that five PgGID1s (A, B, C, E, and F) clustered with AtGID1A and AtGID1C, while the remaining three PgGID1s (D, G, and H) clustered with AtGID1B (Figure 2A). Next, we examined the secondary structure of OsGID1 [41] (Figure 2B) and compared the amino acid sequences of PgGID1s with those of AtGID1s and OsGID1 (Figure 2C). A topology diagram based on the predicted secondary structure of OsGID1 provided the structure information of PgGID1s (Figure 1B). The N-terminal region and two α-helices (α8 and α9; located between the β6 and β7 sheets) of OsGID1 corresponded to the left and right sides of the lid structure, respectively (Figure 2B). It is well characterized that OsGID1, one of the hormone-sensitive lipases (HSLs), contains an evolutionarily conserved HGG sequence and a catalytic triad (S, D, and H), both of which are essential for its enzymatic activity; H in the SDH catalytic triad is replaced by V. In PgGID1s, the V residue is replaced by I, as in AtGID1B and AtGID1C (Figure 2C). Furthermore, PgGID1s showed high sequence similarity with AtGID1s and OsGID1 (Figure 2C). Additionally, important amino acid residues and motifs (Figure 2B) were well-conserved among the PgGIDs, suggesting that PgGID1s function as GA receptors. It is well-known that GID1 proteins act as molecular glue to facilitate the interactions of DELLA repressors with SCF^SLY1/GID2^ in a GA-dependent manner [41]. To confirm subcellular localization of PgGID1s, we co-expressed the *PgGID1-GFP* fusions and *AtARR2-RFP* (as a nuclear marker) in *Arabidopsis* protoplasts. Similar to the nucleocytoplasmic localization of AtGID1s and OsGID1 [42], all four PgGIDs (A–D) localized to the nucleus and cytoplasm (Figure 2D).

### 2.3. Complementation Analysis of the Atgid1a/c Doble Mutant

To evaluate the biological roles of PgGID1s in the GA signaling pathway, we performed complementation analysis of the *Arabidopsis Atgid1a/c* double knockout mutant, which exhibits a GA-deficient semidwarf phenotype [20]. *PgGID1A–D* were individually overexpressed in the *atgid1a/c* double mutant under the control of the constitutively active *35S* promoter (Figure 3A–C). All complementation lines expressing *PgGID1s* showed a wild-type shoot phenotype (Figure 3A) and partially enhanced silique length compared with *Atgid1a/c* plants (Figure 3B). These results indicate that PgGID1s can replace the function of AtGID1s during plant growth and development. Next, we investigated whether exogenous GA_3_ enhances the interaction of PgGIDs with PgDELLA proteins (Figure 3D). We cloned five DELLA-encoding genes of *P. ginseng* (named as *PgRGA1–5*), and ca) (Appendix A), contributing to the essential role for secondary cell wall biosynthesis. We carried out yeast two-hybrid assays in the presence or absence of exogenous GA_3_. In the absence of active GA_3_, rare physical protein interactions between all PgGID1s and PgRGA1, 2 and 3 proteins occurred, but the protein interactions between PgGIDs and PgRGAs (except between PgGID1D and PgRGA1-3) were enhanced in the presence of GA_3_. However, PgGID1D interacted only with PgRGA4 and PgRGA5 in a GA_3_-independent manner (Figure 3D). These results reveal interaction specificity between PgGIDs and PgDELLAs.

### 2.4. Transcriptome Analysis of Root Secondary Growth in Response to GA in P. ginseng

To investigate the root secondary growth of *P. ginseng* in response to GA, we analyzed a total of 5721 genes differentially expressed between DMSO- and GA-treated roots using RNA-seq data (Appendix A). Gene ontology (GO) enrichment analysis of differentially expressed genes (DEGs) revealed significant enrichment of the following functional categories: ‘cell cycle and/or division’ (*p*-values of sub-GO terms in the representative category: *p* = 4.1 × 10^−3^ − 1.6 × 10^−4^), ‘developmental process’ (*p* = 4.7 × 10^−3^ − 3.1 × 10^−8^), ‘cell growth’ (*p* = 1.7 × 10^−3^) and ‘cell wall biogenesis’ (*p* = 9.2 × 10^−3^ − 7.0 × 10^−6^) (Figure 4A). This suggests that GA regulates root secondary growth by controlling root elongation via cell division. Interestingly, the functional enrichment of ‘response to nitrogen’ (*p* = 3.8 × 10^−3^ − 8.6 × 10^−8^), related to nitrate transport and assimilation, was also identified (Figure 4A). Expression of nitrate transporter genes (*NRTs*) was increased, whereas that of nitrate reductase genes (*NIRs*) was decreased, suggesting that GA signaling influences the nitrate metabolic process through the transcriptional regulation of *NRTs* [43]. The results of gene set enrichment analysis (GSEA) further supported the antagonistic transcriptional responses of ABA-related signaling pathway in GA-treated ginseng roots (Figure 4B). Interestingly, JA-responsive genes were significantly enriched among the up-regulated genes in GA-treated *P. ginseng* roots (Figure 4B). These results suggest that JA-mediated downstream signaling pathways are also closely associated with the GA-mediated activation of cambium stem cells.

### 2.5. Cell Wall-Related Genes Play a Major Role in the Secondary Growth of P. ginseng Roots

Next, we focused on the functional enrichment analysis of genes related to ‘plant-type secondary cell-wall biogenesis’ in GA-treated roots. The GO term ‘plant-type secondary cell-wall biogenesis’ was further validated by GSEA using all expressed transcriptome data in *P. ginseng* (adjusted *p*-value = 0.0) (Figure 5A), which showed the most significant enrichment out of analyzed GO terms including cell cycle, cytokinesis, unidimensional cell growth, nitrate assimilation and response to ethylene (Appendix A). In the GSEA, a total of 55 genes were identified to be a critical leading-edge subset of the enriched gene set group leading to enrichment scores with respect to expression changes (Figure 5A; Appendix A). In particular, 18 genes, including *MYB26, GXM1*, *IRX6/9L/12/15L*, *NST1*, *PGSIP3*, *SMB* and *TBL33*, showed significant up-regulation in GA- treated root with more than 2-fold change (*q* < 0.05) (Appendix A), contributing to the essential role for secondary cell wall biosynthesis [44]. Consistently, most of the promoters of GA-responsive genes have GA-responsive elements (GAREs, TAACAAR; Appendix A). This finding suggests that secondary cell wall biogenesis is closely associated with GA-promoted root secondary growth in *P. ginseng*. Consistently, these cell-wall biogenesis-related genes including *WAT1, IRXs, CESAs, NSTs* and *MYB*s were significantly involved in the plant secondary growth by local auxin accumulation [45].

Finally, a transcriptional network of *P**. ginseng* genes related to cell wall biogenesis was analyzed on the basis of homology with *A. thaliana* and its protein-protein interaction network by using STRING database. We found that up-regulated genes in a GA treatment related to cell wall biogenesis showed strong interaction with those related to cell cycle and division, cell growth, response to JA and nitrate assimilation (Figure 5B; Appendix A). Interestingly, genes related to nitrate assimilation were directly associated with ‘cell growth’ and ‘response to JA’ in the gene network analysis (Figure 5B; Appendix A). This suggests that the transcriptional regulatory network affecting GA-induced root secondary growth comprises genes related to the cell cycle, cell division, and cell growth, as well as to the JA response and nitrate assimilation. In this network, genes encoding expansin-like B1 (*EXLB1*), laccase-4 (*IRX12* or *LAC4*), hexosyltransferase (*PGS1P3*), and LRR receptor-like serine/threonine-protein kinase (*FEI1*) were associated with the functions of cell division/cycle, cell growth, or response to JA, thereby playing a crucial role as hub genes in the transcriptional network. Therefore, our results provide biological insight into the transcriptional regulation of root secondary growth in response to GA in *P. ginseng*.

## 3. Discussion

### 3.1. Evolutionarily Conserved GA Signaling Pathways Regulate Root Secondary Growth in P. ginseng

Compared with research on the pharmacological efficacy of *P. ginseng* [3], a limited number of studies has been conducted on the physiology of *P. ginseng,* primarily because of its slow growth habit and difficult cultivation methods. Moreover, functional genomic analysis of plant growth and development of *P. ginseng* has been challenging due to the possible outcomes of functional divergence, which occurs following polyploidization. However, recent advances in genome sequencing technology have enabled the generation of genomic and transcriptomic data from *P. ginseng*. In this study, we successfully identified GA receptor-encoding genes expressed by *P. ginseng* based on whole-genome sequence and transcriptomic data and characterized their physiological responses during root secondary growth. We also carried out functional genomic studies for genetic and physiological regulations of GA (Figure 1). Our results suggest that the secondary growth of *P. ginseng* roots is closely related to the cell cycle/division of cambium stem cells, and to development of starch storing parenchyma cells. Our results also confirmed that GA plays an important role, as a physiological factor, in promoting the secondary growth of *P. ginseng* storage roots.

Since the discovery of the effect of GA on cell elongation and crop yields in plants, the canonical GA signal transduction pathways have been well-characterized in *Arabidopsis* and rice model plant species [15,16,20,46,47]. In this study, we first identified eight GA receptors encoded by the *P. ginseng* genome, and then confirmed their functional conservation using *Arabidopsis* as a heterologous expression system. We showed that four GA receptors of *P. ginseng* (PgGID1s) complemented the *atgd1a/c* double mutant, physically interacted with PgDELLA proteins, and exhibited the same subcellular localization pattern in *Arabidopsis* protoplasts as AtGID1s (Figure 2 and Figure 3). Interestingly, external GA treatment not only enhanced primary shoot growth in *P. ginseng* through cell elongation, but also increased secondary radial growth of the tap root.

### 3.2. GA-induced Root Secondary Growth Is Closely Associated with Cell Wall Biogenesis-Related Network

Transcriptome profiling of *P. ginseng* supported the hypothesis that GA-induced root secondary growth is strongly associated with cell wall biogenesis (Figure 4). The functional role of *GXM1*, *IRX9L/12/15L*, *NST1*, *PGSIP3*, *SMB* and *TBL33* was over-represented with key regulators of secondary cell wall formation in xylem development [48]. Remodeling of the cell wall composition, which is highly flexible and diverse in nature, is important during root growth [44,49]. Similar regulation mechanisms of GA in crop root development, which exogenous GA altered the expression of genes related to cell wall synthesis, have been reported in tobacco [50], sweet potato [36], and carrot [51]. Cambium stem cells in the root undergo rapid division, and new highly dynamic cell walls are formed, resulting in cell elongation [52]. We found that cell wall biogenesis is strongly connected with the functions of ‘cell cycle/division’, ‘cell growth’ and ‘response to JA’ (Figure 5). This result indicates that a transcriptional network regulates root secondary growth in *P. ginseng*. Furthermore, a sub-network associated with nitrate assimilation indirectly interacted with cell wall biogenesis, suggesting that the cell walls in roots readjust to their environment to maximize nutrient availability [53]. In addition, recent studies showed that JA plays a major role in promoting wound healing and stress tolerance through regulation of stem cell homeostasis in plant roots [54]. These results suggest that the assimilation of nitrogen and the regulation of stem cell homeostasis regulated by JA are importantly integrated into the secondary growth of ginseng root. The interaction between the role of inorganic nutrients and hormonal signaling crosstalk to enhance the productivity of root crops will be valuable for future study.

### 3.3. Differential Regulation of the Secondary Growth of P. ginseng Storage Root

In this study, histological staining and microscopic observation of root sections were applied to understand the secondary growth modulated by cambium stem cell activity in *P. ginseng*. Most of the plant secondary growth-related studies have been conducted in *Arabidopsis* and the perennial woody plant species *Betula platyphylla*, with a primary focus on the functions of auxins and cytokinins, which are known to regulate procambium formation [55,56], as well as on stem cell homeostasis and phloem development [27,28,29,57]. On the other hand, studies investigating the role of GA in secondary growth have not been conducted in detail, although it is known that exogeneous GA application increases the number or size of xylem vessels and the lignification of fibers [30,31,32]. The hormonal control of the secondary growth of storage roots has also been documented in annual root crops, including carrot and radish [35,58]. In carrot, exogenous GA application promotes the growth of shoots, while inhibiting the secondary growth of roots. By contrast, in the current study, GA treatment promoted the secondary growth of *P. ginseng* roots by activating cell division and storage parenchyma cell development (Figure 1). These results indicate differences in genetic and physiological factors affecting secondary growth between annual and perennial plants. This is supported by the GA-induced increase in cell division in four different perennial angiosperm trees, including *Fraxinus mandshurica* var. *japonica, Quercus mongolica* var. *grosseserrata, Kalopanax pictus,* and *Populus sieboldii* [59]. These results also suggest that the formation of a signaling network by interaction among plant hormones plays an important role in secondary growth. This further emphasizes the importance of understanding the mechanism of cambium development in perennial plants, in which secondary growth occurs in stages every year.

Understanding the root growth patterns of *P. ginseng* can enhance our knowledge of the growth and developmental mechanisms of perennial plants. Although this study provides substantial evidence supporting the role of GA-induced cell division in the secondary growth of *P. ginseng* roots, the downstream signaling pathways that interact with the upstream GA signaling to regulate storage root development remain unclear. Identification of GA signaling-related genes and molecular mechanisms for its crosstalk with other signaling pathways in perennial plant growth and development are expected to facilitate the breeding and utilization of perennial root crops such as *P. ginseng*.

## 4. Materials and Methods

### 4.1. Plant Materials and Transgenic Plants

One-year-old *P. ginseng* roots (Yunpoong, kindly supplied by the National Institute of Horticultural and Herbal Science) were transplanted into ginseng cultivation soil medium. Two weeks after transplantation of the ginseng seedlings, GA_3_ (Duchefa, Haarlem, The Netherlands), PCZ (Sigma, St. Louis, MO, USA), and DMSO (mock treatment) were treated once a week with a soaking method. After a total of eight treatments, the ginseng roots were sampled, and the main root secondary growth patterns were further analyzed. *Arabidopsis thaliana* Col-0 was used as a wild-type control and *atgid1a/c* double knock out mutants (supplied by RIKEN BRC) were used as the genetic backgrounds of transgenic lines. All plants were grown in a greenhouse under long-day conditions (16-h light/8-h dark cycles) at 22 °C. To generate transgenic plants overexpressing HA-tagged *PgGID1A-D* in the *atgid1a/c* mutant background, the cDNAs were cloned into *pCB302ES* containing the 35S promoter and double HA tag as described previously [60]. All primer sequences for the genes are listed in Appendix A. All transgenes were integrated into the *atgid1a/c* plant genome by Agrobacterium-mediated floral dipping methods with a GV3101 strain. The transgene expression was verified by immunoblotting. Total proteins from 5-days-old seedlings were extracted with protein extraction buffer (50 mM Tris-HCl (pH 7.5), 75 mM NaCl, 5 mM EDTA, 1 mM dithiothreitol, 1× protease inhibitor cocktail (Roche, Basel, Switzerland), and 1% Triton X-100). Total protein was subjected to SDS–PAGE (10% polyacrylamide), transferred to a PVDF membrane and immunodetected with 1/2000 dilution of a peroxidase-conjugated high-affinity anti-HA antibody (Roche, Basel, Switzerland).

### 4.2. Phylogenetic Tree Construction of GA Signaling Related Genes from P. ginseng

The protein sequences of PgGIDs (Genbank accession #: MH050319-MH050322) and PgRGAs (Genbank accession #: MH085925- MH085927) were selected from previous studies [37,38,39]. A phylogenetic tree based on amino acid sequence alignment was generated using MEGA version 7.0 software by the neighbor-joining method with a bootstrap value of 1000 [61]. An online program, TOL (http://itol.embl.de/, accessed on 1 May 2021), was applied to generate the phylogenetic tree. Base on the phylogenetic tree constructed by the GA-related genes from *P. ginseng*, rice and *Arabidopsis*, these genes were divided into different groups and subgroups.

### 4.3. Protoplast Transient Expression Assay and Yeast Two Hybrid Assay

The full-length cDNAs of PgGID1A-D were cloned into plant expression vectors containing HA or GFP tags in the C terminus driven by the 35S:C4PPDK promoter as previously described [60]. For protoplast transient expression assays, about 4 × 10^4^ protoplasts were transfected with 20 μg of plasmid DNA and then incubated under constant light condition at 20 °C for 6 h. For subcellular localization, GFP-tagged constructs were transfected into protoplasts. ARR2-RFP was used as a nuclear marker. GFP and RFP fluorescence were observed with a fluorescence microscope (Nikon). To identify physical interactions between PgGIDs and PgRGAs in the presence of GA_3_, AH109 yeast strains were co-transformed with *pGBKT7*-PgGIDs and *pGADT7-PgRGAs*. Clones showing positive interactions were selected on synthetic medium containing 1 mM 3-aminotriazole (3-AT) without Leu, Trp and His in the presence or absence of 100 nM GA_3_.

### 4.4. Histological Sections and Microscopy

Fresh hand-cut cross sections of *P. ginseng* roots were prepared. Samples were fixed in 1% glutaraldehyde and 4% formaldehyde in PBS pH 7.0 at 4 °C overnight. For paraffin sectioning, tissues were dehydrated (30%, 50%, 70%, 90% and 100% three times, 1 h each), embedded in paraffin, sliced into 10–15 μm-thick sections and mounted onto slides. After dewaxing with Histo-Clear, the slides were dehydrated and counter stained with 1% Safranin-O (Sigma, St. Louis, MO, USA, cat. S2255) and 0.5% Astra blue (Santa-cruz biochem, cat. sc-214558A) for 1 min, rinsed in distilled water, and mounted in Permount mounting medium (Fisher chem., cat. SP15-100, Waltham, MA, USA). The prepared slide samples were observed with bright and polarized light with a Slideview scanner (SLIDEVIEW VS200) and a BX53 microscope (Olympus, Tokyo, Japan). The number of cell rows derived by cambium layers was counted on a straight line traced from the last cell layer of resin ducts to the inner xylem vessel cells.

### 4.5. RNA-seq Analysis

Total RNA was extracted from DMSO- and GA-treated root samples using an Easy Spin RNA Extraction Kit (iNtRON Biotechnology, Seoul, Korea), according to the manufacturer’s instructions. The quality of total RNA was assessed using an Agilent 2100 Bioanalyzer (Agilent Technologies, Santa Clara, CA, USA), and samples with an average RNA integrity number (RIN) of 7.1 and an 28S/18S ratio of 1.0 were selected for RNA-seq analysis. RNA-seq libraries were prepared from 1 μg of total RNA extracted from DMSO- and GA-treated root samples, with three biological replicates per sample, using the TruSeq Stranded mRNA Library Prep Kit (Illumina, Inc., San Diego, CA, USA), according to the manufacturer’s instructions. Then, cDNA was synthesized from mRNA fragments and subjected to end repair, single ‘A’ addition and adapter ligation. The libraries were purified and enriched via PCR amplification, and then sequenced on the Illumina HiSeq 4000 platform to generate 100-bp paired-end reads (Appendix A).

The quality of raw reads was assessed using FastQC (version 0.11.9, Babraham Institute, Cambridge, UK); the quality scores were >Q30, which indicated high quality. Low quality reads (<Q30) and adapter sequences were removed using Trimmomatic [62]. Clean reads obtained from each sample were aligned against the reference genome sequence of *P. ginseng* using HISAT2 [63]. Gene expression was quantified and expressed as fragments per kilobase of transcript per million reads mapped (FPKM) using HTSeq-count [64] and DESeq2 [65]. Genes differentially expressed between each replicate of DMSO- and GA-treated root samples were identified using DESeq2 [65], based on cutoff values of *q* < 0.05 and FC ≥ 1.5. GO enrichment analysis of DEGs homologous to Arabidopsis genes (TAIR 10 release) was performed using DAVID, with an *EASE* score cutoff of <0.01. Genes enriched under specific GO terms were selected, and their expression patterns were visualized as a heatmap using MeV (http://mev.tm4.org, accessed on 21 March 2021). Expression levels of genes were shown as Z-score of FPKM values. To test if a particular gene set (i.e., secondary cell wall biogenesis) was enriched, GSEA [66] (version 4.0.3, Broad Institute of Massachusetts Institute of Technology and Harvard, Cambridge, MA, USA) was applied to a background dataset comprising transcripts with FPKM > 0.3, which balances the numbers of false positives and false negatives, in either DMSO- or GA-treated root samples. To investigate protein–protein interactions, DEGs, especially up-regulated genes, categorized with specific GO terms were searched against STRING [67], with medium confidence (≥0.4). The network was further analyzed using Cytoscape (version 3.7.1, Cytoscape Consortium) based on the degree of connectivity among nodes.

## Figures and Tables

**Figure 1 ijms-22-08694-f001:**
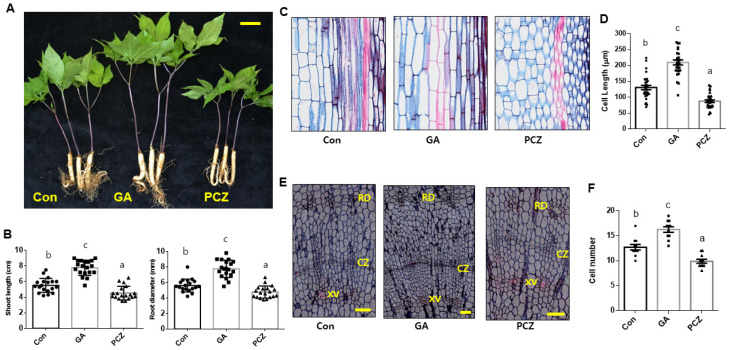
Exogenous gibberellin (GA) treatment promotes primary growth of stems and secondary growth of roots in *Panax ginseng*. (**A**) Phenotype of 1-year-old *P. ginseng* plants treated with DMSO (control [Con]), 10 μM GA_3_, and 100 μM paclobutrazol (PCZ) once a week for 8 weeks. Scale bar = 2 cm. (**B**) Measurements of shoot length and root diameter. (**C**) Representative images of stained stem cross-sections of *P. ginseng* plants treated with DMSO (Con), GA_3_ and PCZ. Scale bar = 100 μm. (**D**) Quantification of cell length in the indicated treatments. (**E**) Representative images of stained root cross-sections of *P. ginseng* plants treated with DMSO (Con), GA_3_, and PCZ. XV: xylem vessel, CZ: cambial cell layer zone, RD: resin duct cells. Scale bar = 100 μm (**F**) Quantification of cambium-derived cells in the XV and RD of each ray. In (**B**,**D**,**F**), dots, squares and triangles represent individual values. Error bars represent standard error; *n* = 16 (**B**), 20 (**D**), 10 (**F**). Different lowercase letters indicate statistically significant differences (*p* < 0.05; one-way analysis of variance [ANOVA], followed by Tukey’s multiple range test).

**Figure 2 ijms-22-08694-f002:**
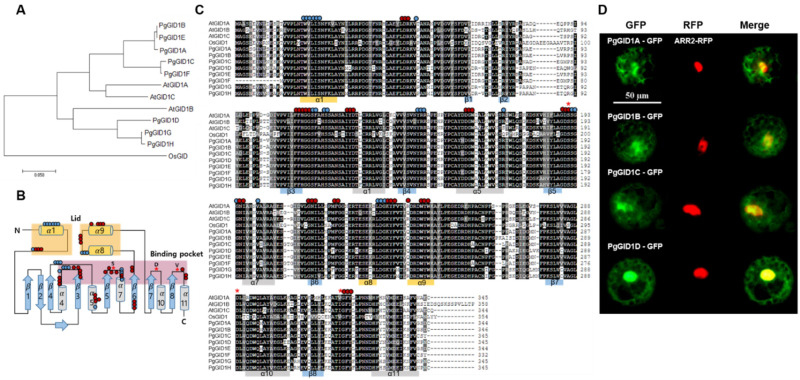
Phylogenetic analysis and amino acid sequence alignment of GID1 proteins. (**A**) Phylogenetic analysis of PgGID1A–H, AtGID1s, and OsGID1. The phylogenetic tree was constructed using the MEGA7 program. Horizontal branch lengths are proportional to the estimated number of amino acid substitutions per residue. Bootstrap values were obtained by 1000 bootstrap replicates. Pg, *Panax ginseng*; At, *Arabidopsis thaliana*; Os, *Oryza sativa*. (**B**) Topology diagram based on the predicted secondary structure of the OsGID1 protein [15]. (Blue circles indicate important residues involved in the GID1–SLR1 interaction, and red dots indicate important residues involved in OsGID1–GA and GID1–SLR1 interactions. Red stars indicate residues essential for enzymatic activity. Colored zones indicate the lid (yellow) and binding pocket (red). (**C**) Amino acid sequence alignment of the GID1 proteins of *Arabidopsis*, rice, and *P. ginseng* constructed using SMS (https://www.Bioinformatics.org (accessed on 1 May 2021)). (**D**) Subcellular localization analysis of PgGID1A–D proteins in Arabidopsis protoplasts. Full-length coding sequences of *PgGID1A–D* were fused to the *GFP* reporter gene. The nucleus was visualized using the AtARR2-RFP nuclear marker. GFP and RFP fluorescence images were merged. Scale bar = 50 μm.

**Figure 3 ijms-22-08694-f003:**
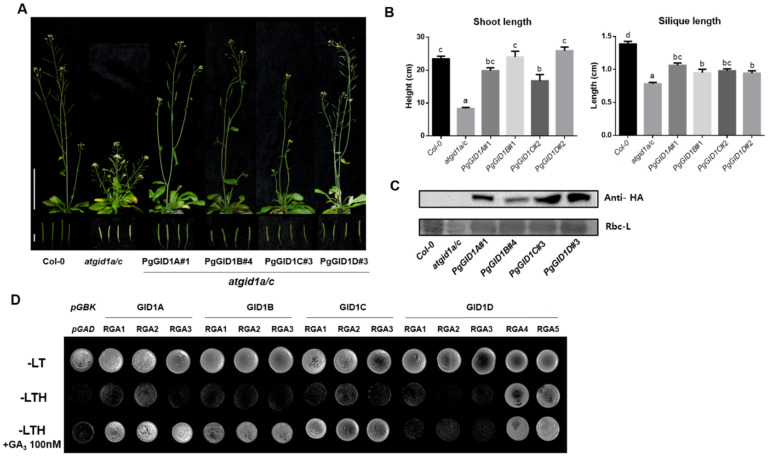
Complementation of the Arabidopsis *gid1* loss-of-function mutant via *PgGID1* overexpression. (**A**) Rescue of the GA-insensitive dwarf phenotype of the *atgid1a/c* double mutant by overexpression of *PgGID1A–D* genes. Scale bar = 5 cm (**top** panel) and 0.5 cm (**bottom** panel). (**B**) Measurement of the length of shoots and siliques shown in (**A**). Error bars represent the standard error (*n* > 12). Different lowercase letters indicate statistically significant differences (*p* < 0.05; one-way ANOVA, followed by Tukey’s multiple range test). (**C**) Western blot analysis of PgGID1A–D protein levels in *atgid1a/c* plants. (**D**) Yeast two-hybrid assay. To test the interaction between PgGID1s and PgRGAs, *PgGID1A–D*-carrying pGBKT7 constructs (bait) were co-expressed with *PgRGA1–5*-harboring pGADT7 constructs (prey) in yeast cells, which were grown on media (-LTH) supplemented with or without 100 nM GA_3_.

**Figure 4 ijms-22-08694-f004:**
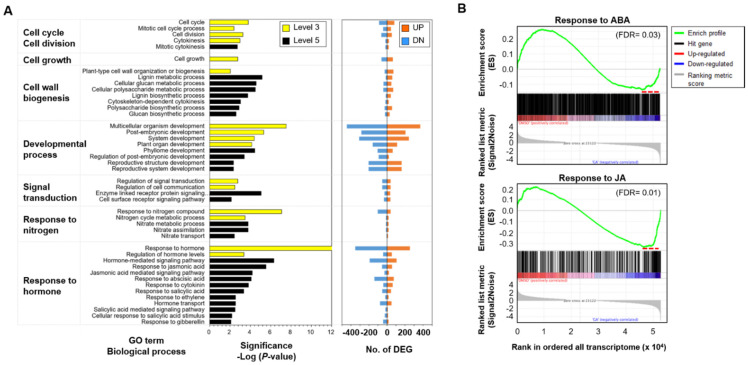
Transcriptome profiling of *P. ginseng* roots treated with or without GA. (**A**) Gene ontology (GO) enrichment analysis of differentially expressed genes (DEGs) identified by comparison of GA- and DMSO-treated root samples. GO terms of level 3 (yellow bars) and level 5 (black bars), with EASE score < 0.01, were selected (left panel). The number of up-regulated genes (red) and down-regulated genes (blue) categorized under the enriched GO terms are shown in the right panel. (**B**) Enrichment plot for the responses to ABA (GO: 0009737) and JA (GO: 0009753) in the transcriptome data of DMSO- and GA-treated root samples. In the enrichment plot, the red dotted line represents the gene subset that made the largest contribution to the enrichment score (ES) (false discovery rate [FDR] < 0.05). The ranking list metric in the plot measures the correlation between a gene and the plant phenotype. In the ranking list, positive values indicate genes up-regulated in DMSO-treated root samples with red color gradient, and negative values indicate genes down-regulated in DMSO-treated root samples.

**Figure 5 ijms-22-08694-f005:**
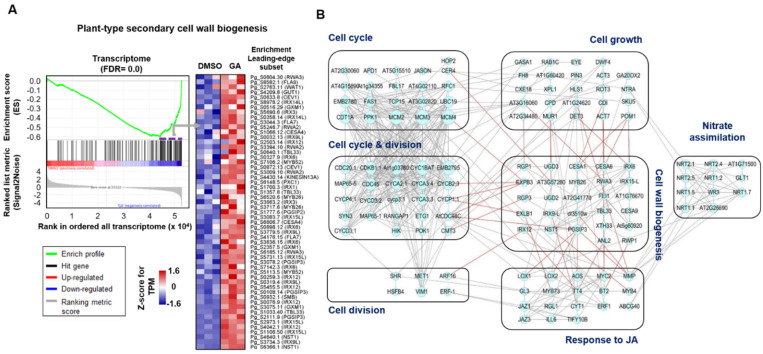
Functional enrichment of cell wall biogenesis in *P. ginseng* root in response to GA and the function-related transcriptional network. (**A**) Enrichment plot for the plant-type secondary cell wall biogenesis (GO: 0009834), and an expression heatmap of 55 genes related to this pathway (FDR = 0.0). (**B**) Transcriptional network of up-regulated genes related to cell wall biogenesis (57 genes), cell cycle and division (120 genes), cell growth (54 genes), response to JA (42 genes), and nitrate assimilation (13 genes). In the network, red lines indicate the connection of genes between cell wall biogenesis and other functional categories.

## Data Availability

The Raw data were deposited in the NCBI Short Read Archive database under the accession number; SAMN12240097 for GA-treated *P. ginseng* root and SAMN122731128 for DMSO-treated *P. ginseng* root as Control.

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
