# Peer review of "Gibberellin Signaling Promotes the Secondary Growth of Storage Roots in Panax ginseng"

_ijms, 2021, doi:10.3390/ijms22168694_

Round 1

Reviewer 1 Report

Hong et al.

The paper is a basic investigation on Ginseng root growth by applying mostly genomic/ transcriptomic methods. The results are clear and the paper is well-written. However, the figures are all too small. Have you ever tried to read your printout? I also would not put all figures together after the text but each text part belonging to one figure in front of that figure. The figures must be reformatted so that one can read what is on them, probably twice as big in general with special care on the size of letters etc.

You concentrate on tap root growth but what about the content of secondary metabolites making Ginseng such avaluable plant. Bigger roots may be better for yield of medically valuable metabolites or not.

Author Response

Comment 1.

The paper is a basic investigation on Ginseng root growth by applying mostly genomic/ transcriptomic methods. The results are clear and the paper is well-written. However, the figures are all too small. Have you ever tried to read your printout? I also would not put all figures together after the text but each text part belonging to one figure in front of that figure. The figures must be reformatted so that one can read what is on them, probably twice as big in general with special care on the size of letters etc.

Our response : We are very sorry for these mistakes. We addressed these issues as much as possible in the revised manuscript.

Comment 2.

You concentrate on tap root growth but what about the content of secondary metabolites making Ginseng such a valuable plant. Bigger roots may be better for yield of medically valuable metabolites or not.

 Our response : Thank you for this valuable comment. In this study, we focused on the physiological role of GA signaling in storage root secondary growth of P. ginseng through exogenous GA effects and transcriptome analysis. As expected, we tried to confirm the change in the transcriptional response in genes related to the biosynthesis of secondary metabolites such as terpenoids and saponins. Interestingly, the terpenoid synthesis pathway was positively correlated in GA-treated ginseng roots, supposing that GA treatment would affect to the biosynthesis of medically valuable metabolites. We are currently attempting to measure saponin content in GA-treated ginseng tissue, but these interesting results are not relevant to the main topic of the current version of the manuscript. So, we decided to submit this interesting result to the next prepared manuscript.

Reviewer 2 Report

In this work, the authors take advantage of recent genome sequence information in P. ginseng to examine the role of gibberellins (GAs) in development and evolutionary conservation of the genes involved in response to GA.

The authors first show that application of GA enhances growth of stems and the tap root, while the GA biosynthetic inhibitor PCZ had the opposite effect. At the cellular level, the stem growth correlates with elongated epidermal cells, while in the root, it is through increased parenchyma cells.

Next, the authors show that the P. ginseng genome encodes 8 putative GA receptors through sequence homology with GID1 from Arabidopsis; these contain structural motifs are found in common with GID1s from Arabidopsis and Oryza (rice). The high number of orthologues is consistent with whole-genome duplication events in the evolution of P. ginseng. Four of the ginseng GID orthologues show expression by GFP reporter in the nucleus and cytoplasm in Arabidopsis protoplasts. To test for functional conservation, the authors test the ability of 35S promoter-driven constructions for these four GID orthologues to individually complement the loss-of-function phenotype of Arabidopsis atgid1a/c double mutants. They find that shoot length is restored to near wild-type levels, while silique length is partially rescued. Using yeast two-hybrid analysis, they further show that the presence of exogenous gibberellins increases interaction of ginseng GID with putative ginseng DELLA proteins, which are predicted to interact with GID in a GA dependent manner from work in other plant systems. They find that most of these interactions do occur, consistent with conserved function. GID1D showed interaction with only RGA4 and RGA5, but not RGA1/2/3, which was also independent of exogenous GA. This suggests that the GID1-RGA interactions have some specificity.

The authors next perform RNA-Seq analysis on root tissue to analyze gene expression differences in control and +GA conditions. They find that genes in the cell cycle and development are enriched. GA also seems to enhance nitrate transport gene expression, and genes responsive to JA were also enriched suggesting some pathway similarities in P. ginseng. These associations were further validated bioinformatically by examining differential expression of cell-wall biogenesis genes and using orthology with Arabidopsis protein-protein networks to predict similar networks in ginseng.

The work is solid and the presentation is straightforward and well presented, and makes an advance in the understanding of evolution of gibberellin function and genetics in P. ginseng compared with the models Arabidopsis and Oryza. The supplementary data seem appropriate, in particular the results from replicates of the control and +GA transcriptomics results.

Major comments

Is there a reason why the P. ginseng GID1D had such a different profile of interaction with the RGA orthologues? Was there something in the structures that predicted this?

Do promoters of the genes that are responsive to exogenous GA show expected cis-regulatory motifs?

Minor comments

line 2, title: 'ginseng' be in lower case as the species name

line 108 – correct spelling of P. ginseng

line 169, a word missing between 'whether' and 'enhances' – should this be 'exogenous gibberellins' or 'GA3'?

Author Response

Comment 1.

Is there a reason why the P. ginseng GID1D had such a different profile of interaction with the RGA orthologues? Was there something in the structures that predicted this?

Our response : Thank you for these critical comments. To properly respond to these comments, we tried to identify any significant differences between PgGID1D and others in primary and secondary structures, but there are no significant differences in evolutionarily conserved motifs. Next, 3D structural models were predicted using PHYRE II analysis and PyMOL structural visualization software. As presented below, the predicted 3D structures could not provide any significant differences to account for the different profiles of interactions with RGA orthologs. We believe that these differences are probably caused by differences due to different non-conserved amino acid sequences. This difference will provide GA signaling specificity during growth and development of ginseng. 

Comment 2.

Do promoters of the genes that are responsive to exogenous GA show expected cis-regulatory motifs?

Our response : Thank you for this critical comment. Based on the reviewer's suggestion, we identified for GA-responsive cis-regulatory elements (GAREs) in the promoters of GA-responsive genes in P. ginseng. As newly provided in Table S3, 32 of the 54 GA-responsive genes shown in Figure 5A have putative GARE motifs.

Minor comments

line 2, title: 'ginseng' be in lower case as the species name

line 108 – correct spelling of P. ginseng

line 169, a word missing between 'whether' and 'enhances' – should this be 'exogenous gibberellins' or 'GA3'?

Our response : We are very sorry for these minor points. We addressed these minor errors in the revised manuscript.

Round 2

Reviewer 2 Report

The minor revisions were incorporated and look fine. A minor comment, line 108 needs a capital 'P' in P. ginseng. Otherwise good for publication.